# Effects of Cheliped Amputation on the Personality of Crayfish

**DOI:** 10.3390/ani14071132

**Published:** 2024-04-08

**Authors:** Leiyu Lu, Li Su, Mengdi Si, Guangyao Wang, Chunlin Li

**Affiliations:** 1School of Resources and Environmental Engineering, Anhui University, Hefei 230601, China; luleiyu1949@163.com (L.L.); suli021777@163.com (L.S.); wgy19980914@163.com (G.W.); 2Anhui Province Key Laboratory of Wetland Ecosystem Protection and Restoration, Anhui University, Hefei 230601, China

**Keywords:** animal personality, behavioral syndromes, repeatability, aggression, cheliped amputation, crayfish

## Abstract

**Simple Summary:**

Morphological changes in early life stages may have a strong influence on animal personalities in adulthood, which describe inter-individual differences and intra-individual consistency in behaviors across time and contexts; thus, investigating this relationship can shed light on personality development throughout ontogeny. This study examined juvenile crayfish reared with different degrees of cheliped mutilation and explored their personality patterns, including exploration and aggression. Our findings reveal that personality differences in adult crayfish may be influenced by the extent of cheliped mutilation during the crayfish’s growth. Crayfish had high repeatability in exploration and aggression, with males having higher repeatability than females. Our study would help in better understanding the role of morphological mutilations in the development of animal personalities.

**Abstract:**

Animal personality, which describes inter-individual differences and intra-individual consistency in behaviors across time and contexts, has been widely observed and has significance for both ecology and evolution. Morphological modifications, particularly during early life stages, may highly influence animal behavior in adulthood; thus, exploring this relationship can elucidate personality development throughout ontogeny. In this study, we reared juvenile crayfish (*Procambarus clarkii*) with different degrees of cheliped mutilation and explored their personality patterns, including exploration and aggression, when they reached sexual maturity. Male crayfish showed repeatability in exploration, and both sexes showed repeatability in aggression. We observed no significant correlation between the two behavioral traits, indicating the absence of behavioral syndromes. Moreover, exploration did not differ according to the type of mutilation, but crayfish with more intact chelipeds were more aggressive, and males were more aggressive than females. These results indicate that cheliped mutilation may modify the average levels of personality traits associated with competition or self-defense. Our study provides insights into how morphological modifications may shape animal personalities in adulthood.

## 1. Introduction

Animal personality refers to consistent behavioral differences observed among individuals over time and across different situations [1]. Animal personality is common [2,3] and contributes to intra-specific variation, which may have important ecological and evolutionary implications [4]. Personalities affect the resource use and response to risk of animals, leading to differences in their ecological niche [5]. Furthermore, personality influences the degree to which individuals can adapt to their environment and modify their life history strategies [6]. Investigating the ontogeny of personality can shed light on the underlying mechanisms responsible for inter-individual behavioral differences.

Animal personalities may be shaped by genetic background [7] and multiple internal and external factors [8]. Previous studies have found that individuals raised in different environments, even those with the same genotype, may differ with respect to personality type [9]. Internal and external environmental factors have mixed effects on the ontogeny of personality. For example, zebrafish (*Danio rerio*) with longer bodies are less inclined to explore [10]. Environmental complexity during early life may reduce the levels of boldness and exploration in mosquitofish (*Gambusia affinis*) [11]. Furthermore, aggression in Micronesian starling (*Aplonis opaca*) parents increases with the age of their offspring [12]. Despite a wealth of research investigating the effects of environmental factors on animal personality, consensus has not yet been achieved because of the complex underlying mechanisms [13]. Additionally, few studies have related animal personality to modifications in morphological characteristics that are strongly linked to behavior.

Natural selection leads to the evolution of morphological traits necessary for important behaviors, such as foraging and reproduction [14,15]. The proper functioning of these behaviors relies on an intact morphological structure. However, various deterministic or stochastic events throughout an animal’s life, such as physical damage, can modify its morphological characteristics [16,17]. Given the strong link between morphology and behavior, changes in these two phenotypic traits are expected to interact [18]. For example, alterations in tooth morphology can influence animal dietary preferences with age [19]. Additionally, animals with active and energetic personalities may exhibit smaller body sizes at sexual maturity, demonstrating the effect of personality on shaping morphological characteristics [20,21]. Body mutilation can reduce access to resources or hinder success in intra- and inter-specific interactions, causing animals to adjust their behavioral strategies. For example, crustaceans with incomplete chelipeds are less likely to win fights or obtain food resources, leading to fewer opportunities for reproductive success [22]. Given that morphological alterations are common throughout an animal’s lifetime, exploring their effects on behavior can provide valuable insights into behavioral development.

Behavioral ontogeny is a long-term process, and the effects of behavioral factors differ at different developmental stages [23,24]. The early life stage is commonly considered a key “sensitive window” during which behavioral traits are expected to be strongly shaped by experience [25]. Growing up in certain environments can induce animals to exhibit permanent personality traits. Therefore, studies exploring the effects of these factors on animal personality should focus on the sensitive early stages. Owing to their weak defense capacity at this stage, animals often get hurt, which can result in body mutilation at a young age. Such physical defects, particularly those related to resource finding and intra- and inter-specific interactions, may influence subsequent behavioral development [26]. Thus, exploring how body mutilations during early life affect personality may elucidate animal behavioral adaptations to their environments.

*Procambarus clarkii* is a freshwater crustacean that belongs to the Cambaridae family. Both male and female crayfish exhibit significant secondary sexual characteristics. Female crayfish have a genital opening located at the base of the third pereiopod, with a pair of dark, circular holes visible, whereas male crayfish have a pair of hooks located inside the fifth pair of pereiopods. In crustaceans, the cheliped is a multifunctional organ that combines offense, defense, and predation [27]. Crayfish chelipeds are composed of movable components called dactyls and a fixed part called the propodus. The junction connecting the dactyl and propodus is referred to as the joint. Breakage of the chelipeds in the event of danger or injury is a common phenomenon in crustaceans [28]. Individuals with amputated chelipeds are subsequently more vulnerable to attacks by other individuals in a group [27]; cheliped mutilation also reduces their ability to kill prey and explore the environment for shelter [29,30]. However, it remains unclear whether cheliped mutilation in crayfish causes personality changes that lead to altered intra- and inter-specific relationships.

In this study, we investigated juvenile crayfish reared with different levels of cheliped mutilation: severe mutilation (SM; one cheliped was entirely removed), medium mutilation (MM; the mobile dactyl of one cheliped was removed), and no mutilation (NM; both chelipeds remained intact). Upon sexual maturity, we measured and compared two behavioral traits, exploration and aggression, among the three treatments and determined behavioral repeatability and correlations. Considering the wide presence of personality traits in the animal kingdom [3,31,32], the exploration and aggression of crayfish may have been repeatable in each treatment. However, there may have been no correlation between these two behavioral traits because aggression is predominantly associated with resource defense, whereas exploration is associated with resource finding [33,34]. Moreover, crayfish must explore the environment for their resource needs, regardless of cheliped intactness; thus, we expected no differences in exploration among the treatments [35]. However, cheliped intactness largely determines the results of intra- and inter-specific competition [36], particularly among males; therefore, we predicted that males would be more aggressive than females and that crayfish with intact chelipeds would exhibit higher levels of aggression. The results of this study may help us understand the role of morphological mutilations during early life in shaping animal personalities.

## 2. Materials and Methods

### 2.1. Ethical Note

The experiments complied with the current animal welfare and scientific research ethics legislation in China. All animal care and experimental procedures were approved by the Institutional Animal Care and Use Committee of Anhui University (permission no. 2023-070). At the end of the study, the crayfish were kept in laboratory tanks for other behavioral studies.

### 2.2. Experimental Animals and Rearing Conditions

The juvenile crayfish used in this study were procured from Hefei Huanghua Market (Co, Hefei, China) and reared in the laboratory at Anhui University (117.18° E, 31.77° N). Because these crayfish were randomly captured from large breeding ponds in which newborns birthed by a large number of females were being reared, the influence of genetic relatedness among individuals could be neglected. After acclimatization to the laboratory rearing environment, we conducted active assessments of the juvenile crayfish housed in the tanks and excluded inactive individuals from the experiment. Finally, we obtained 180 juvenile crayfish that satisfied the experimental conditions. Following endorsement from the Animal Care and Use Committee of Anhui University (IACUC, AHU), the chelipeds of the crayfish were cut with surgical scissors to different degrees as follows: severe mutilation (SM, one cheliped was entirely removed by cutting at the base), medium mutilation (MM, the mobile dyctyl of one cheliped was removed), and no mutilation (NM, both chelipeds remained intact). To ensure proper sanitary conditions, we disinfected the wounds of individuals subjected to mutilation using iodophor. A natural light cycle of 14:10 h (light/dark) was maintained throughout the experiment.

We segregated the 180 juvenile crayfish into 18 small water tanks according to the integrity of their chelipeds and their sex. The 18 small water tanks were designated with unique identification numbers (blocks) and categorized into three groups according to the type of treatment (SM, MM, and NM). Each group consisted of six small water tanks containing five female and five male juvenile crayfish with comparable cheliped integrity. The water depth in each small tank was 8 cm, and the dissolved oxygen, pH, and water temperature were set to >5.0 mg/L, 7.0–8.5, and (25 ± 1) °C, respectively. During early rearing, the subjects were fed fresh vegetable leaves, corn, and other plant-based baits together with a small amount of live animal-based feed sourced from the flour weevil *Tenebrio molitor.* After one week, the animals were fed commercial lobster puffed feed (Yangzhou HONGDA FEED Co., Ltd., Yangzhou, China) containing ≥32% crude protein, ≥8% crude fiber, ≥4% crude fat, ≥15% crude ash, ≥1.5% lysine, ≥1% total phosphorus, 0.5–3.0% calcium, and 0.5–2.5% sodium chloride. The crayfish were fed twice daily, at 8:30 and 18:30. Additionally, 50% of the water in the small tanks was changed every week to maintain water quality.

Changes in body color are an outward indication of the degree of maturity of crayfish, which are usually greenish in color before sexual maturity and dark red after sexual maturity [37]. For crayfish that had regenerated their cheliped during rearing, the researchers cut them again and disinfected the wounds. After three months of rearing the juvenile crayfish, all crayfish were examined to determine whether they had reached sexual maturity. A predetermined criterion was used 12 h prior to the experiment to select the tank for testing the following day. We refrained from feeding the crayfish residing in the experimental tank, whereas the small tanks were kept under natural photoperiod conditions throughout the course of the experiment. Prior to the experiment, we identified and marked the backs of selected crayfish using a marker. After completing the experiment, the crayfish were returned to their original small tanks to maintain habitat consistency across all three experiments and ensure that individual IDs remained unchanged for all subsequent behavioral experiments.

### 2.3. Exploration Test

We conducted the exploration experiment in a blue opaque rectangular container (60 cm long × 37.5 cm wide × 15 cm high), which was designed with a white opaque enclosure fixed at one end. This enclosure comprised a 15 cm × 15 cm square shelter with a movable trap door (15 cm long × 15 cm wide) connected to a fishing line (Figure 1). A proficient experimenter gently opened the movable door of the shelter to allow the crayfish to crawl out. To create a more complex environment for the crayfish, we randomly placed obstructions inside the container, thus hindering the animals’ view of their surroundings. Throughout the experiment, a video camera (Sony HDR-CX510, 55 Extended Zoom; Sony Corporation, Tokyo, Japan) was suspended above the center of the experimental tank to record crayfish behavior. The experimental blue plastic box was filled with dechlorinated tap water to a depth of 8 cm. Between each experiment, the water was replaced to prevent the effects of odors and chemical signals released by previously tested subjects on subsequent test subjects. During the experiment, an opaque curtain was used to conceal the observer and minimize disturbance to the test subject. To ensure that individuals did not become habituated to the experimental items, we used different items in the three replicate experiments. Accordingly, broken porcelain blocks, small solid black boxes, and simulated leaves were placed in identically arranged test chambers.

During the experiment, we randomly chose subjects and gently placed them in a closed shelter before turning on the camera. Each tested individual was allowed a 5 min acclimation period, following which the experimenter carefully pulled open the flap door of the shelter remotely and kept it open for the duration of the experiment. We considered the experiment to have started once the whole body of the crayfish had emerged from the shelter. We continued to record the movements of the subjects for 10 min using a camera (exploration analysis). After the test, the experimental individuals were immediately transferred back to their previous aquaculture tanks to ensure consistency in their living environment. Each crayfish was tested in triplicate. The first test was conducted in a random order, whereas the same order as the first test was followed for the subsequent two tests. Furthermore, we ensured a one-week interval between any two consecutive tests to avoid the influence of short intervals on the experimental results. From each 10 min motion video (one frame per second), we extracted 600 images and used ImageJ software (ImageJ v.220706; http://rsbweb.nih.gov/ij/; accessed on 12 October 2022) to mark the position of the crayfish head in each frame and document its motion path. We quantified the exploration score of the focal subject using the total path length, similar to the approach used by Eden brow and Croft [38]. We defined exploration as crayfish moving around with their antennae and chelipeds touching the surroundings, and we recognize that the measurement of exploration cannot be precisely distinguished from that of activity because exploration involves being active [39]. Nonetheless, we referred to this test as an exploration task because it was conducted in an unfamiliar environment, and the distance covered within this environment was accepted as an exploration metric [4,40], despite the possibility of incorporating concurrent activities.

### 2.4. Aggression Test

We used a cylindrical, opaque plastic bucket with an anti-skid cloth adhered to the bottom to measure aggression. In this aggression experiment, we introduced a rubber crayfish model and forceps to simulate situations in which crayfish were attacked by conspecifics. The crayfish model was secured to the front end of a thin wooden stick, which enabled the experimenter to control the stick and vary the intensity of the attack on the individual.

We categorized the crayfish attacks into three levels: (I) non-contact, which involved rapid and vertical descent of the crayfish model from a fixed height (10 cm) to within 1 cm of the crayfish cephalothorax; (II) contact, which involved rapid and vertical descent until the crayfish model made contact with the crayfish cephalothorax; and (III) vertical clamping of the crayfish cephalothorax using forceps to immobilize the crayfish. The attack levels were progressively increased for each crayfish.

We randomly selected experimental individuals and allowed them a 5 min adaptation period in the arena to acclimate to the experimental environment. We then performed aggression experiments at the three levels in order, with 10 attacks at each level conducted by the experimenter. The interval between attacks at each level was approximately 5 s, whereas that between attacks at different levels was 10 s. We repeated the aggression experiment three times, with a seven-day interval between sessions.

The criteria for evaluating aggression were as follows: experimental individuals were scored 10 points if they raised their chelipeds during a non-contact attack, whereas failing to do so resulted in zero points. The behavior of raising chelipeds during the contact attack earned five points, whereas not doing so resulted in zero points. When using tweezers to clamp the crayfish’s back, 2.5 points were attributed to individuals that lifted their chelipeds, and zero points were awarded to those who did not. We then calculated the total score for each crayfish after conducting 10 experiments at each attack level (I, II, and III). When crayfish raised their chelipeds in all 30 experiments, they received a perfect score of 175.

### 2.5. Statistical Analyses

Repeatability, which considers individual identity as a grouping factor, is often employed as a measure of behavioral consistency within individuals [41]. In this study, we used the *rpt* function from the R package (R v.3.6.3) *rptR* [41] to assess repeatability in aggression and exploration, using block ID and individual ID as random effects. We set the Nboot argument to 1000 bootstraps to control the number of confidence interval estimates for parameter bootstrap iterations. Additionally, we used the corr.test function in the *psych* package [42] to measure the Spearman’s rank correlation between the two behaviors.

To examine the two behavioral features, we conducted the Shapiro–Wilk test to assess the normality assumption. We then fitted a generalized linear model with a Gaussian error structure to test the impact of cheliped integrity and sex on each behavior. We used the emmeans package to compare cheliped integrity by sex after fitting the model. We initially included the interaction between cheliped integrity and sex in the model; however, this was subsequently removed owing to a lack of statistical significance. All statistical analyses were performed using R version 4.2.0 (TEAM, 2009), with a significance level of *p* < 0.05 established for all tests.

## 3. Results

The mean degree of exploration and aggression scores according to crayfish sex and degree of mutilation are shown in Table 1. Males with broken and intact chelipeds displayed significant repeatability during the behavioral tests (Table 2 and Table 3). All crayfish exhibited significant repeatability in aggression (Table 3). We observed no significant correlation between aggression and exploratory behavior traits in any individuals (Figure 2). Moreover, our findings suggest that neither sex nor cheliped integrity had a significant effect on the experimental groups’ exploration (Figure 3). However, experimental groups with higher cheliped integrity demonstrated greater aggression (Figure 3). Additionally, males exhibited higher levels of aggression than females (Figure 3).

## 4. Discussion

Repeatability in exploration and aggression in individual crayfish provides evidence for the existence of individuality in invertebrates, further suggesting that animal personalities are widespread throughout the animal kingdom [43]. The existence of an animal personality can change the direction of group evolution and effects of natural selection [44]. Animal personality may also lead to differences in the living spaces of different individuals and further enhance differences in physiological and ecological characteristics between individuals [45]. Changes in animal morphological characteristics appear to affect behavior; thus, our research enhances our understanding of animal behavior and may contribute to a wider acceptance of animal personalities [46].

Behavior repeatability helps animals gain more benefits in their natural environment [47], and our study found widespread repeatability in exploration and aggression (Table 2). In addition, our results showed that behavioral repeatability was higher in males than in females, which is consistent with previous studies [48,49]. Higher behavioral repeatability in males can help them attract the opposite sex and obtain more reproductive opportunities [50]. We also found that crayfish exhibited higher repeatability in aggression than in exploration (Table 2). Previous studies have shown that aggression is more reproducible than other behaviors, such as activity and migration [51]. Repeatability in aggression helps animals occupy more territory, thereby ensuring better access to resources and increased reproductive success [52,53]. High repeatability in aggression has been estimated in both artificial and wild populations [54,55].

In addition to repeatability, animal personality also includes correlations between behaviors [39]. Behavioral correlation is also a combination of related behaviors in different situations, such as behavioral syndromes in behavioral ecology research [56]. Behavioral syndromes enable animals to exhibit the most adaptive combination of behaviors and gain significant benefits in obtaining various resources for survival [57]. In this study, no correlation was observed between aggression and exploration (Figure 2). This may be because aggression is driven more by hormonal stimuli, external stimuli, and fear of predators [58]. In contrast, exploration was a hunger-driven behavior, hence the lack of correlation between behaviors. According to current research, the correlation between behaviors is inconsistent [59]. Various ecological factors, such as low predation pressure, may affect the results of correlations between behaviors, as it is difficult to detect correlations under low predation pressure [60]. Additionally, some studies have suggested that the relevance of exploration as a behavioral trait may not be easily generalized across different species [61].

We observed no significant difference in the exploratory abilities of crayfish with different degrees of mutilation (Figure 3). First, exploration is driven by resource needs [62]; in our experiments, crayfish exploration was food-driven. Regardless of the degree of mutilation, crayfish must explore their environment to obtain food for survival. Second, the risk of predation during foraging can affect exploratory behavior [63]. In our experiments, exploration behavior was less likely to differ at the same or a lower risk of predation. As expected, individuals with higher cheliped integrity exhibited higher levels of aggression (Figure 3). An increased degree of cheliped mutilation led to a reduction in aggression, suggesting that changes in morphological structure can lead to changes in animal behavior [64]. This may be because the cheliped has evolved as a key weapon of attack and defense for crustaceans and breakage of the cheliped causes crayfish to become less capable of attack and defense [36]. In this case, crayfish reduce aggression to reduce their risk of death and ensure their survival [65,66].

Furthermore, our results did not reflect differences in exploration between crayfish sexes; however, male crayfish were more aggressive than female crayfish. Previous research has shown that individuals with greater competitive abilities may be more exploratory [67]. Males also exhibit more competitiveness than females [68]. However, research on the effect of sex on exploration behavior remains controversial, with some studies suggesting that females may explore less and others showing opposite results [11,69]. This suggests that sex may interact with other factors to influence exploration [70,71]. In general, sex differences play a crucial role in aggressive interactions among adult crayfish [72,73]. Crayfish use aggression to gain a dominant position in the population for better resources and more opportunities to reproduce [74]. Previous studies have shown that adult male crayfish occupy a higher status and rank in the population than adult female crayfish [72,75]. Therefore, males exhibit higher levels of aggression to gain dominance within the group [76,77].

## 5. Conclusions

Our study reveals the presence of personality traits. Repeatability in exploration and aggression was widespread in crayfish, with males exhibiting overall higher repeatability than females. Aggression and exploration were not correlated; however, such a correlation is affected by various factors, suggesting that further studies are required to reveal the underlying mechanism. Crayfish with different degrees of cheliped mutilation showed no differences in exploration; however, individuals with less cheliped mutilation exhibited higher levels of aggression. Moreover, male crayfish were more aggressive than female crayfish; however, we observed no effect of sex on exploration. In conclusion, our study provides evidence for the existence of personality traits in invertebrates and suggests that the degree of cheliped mutilation during crayfish growth and development contributes to personality differences in adults.

## Figures and Tables

**Figure 1 animals-14-01132-f001:**
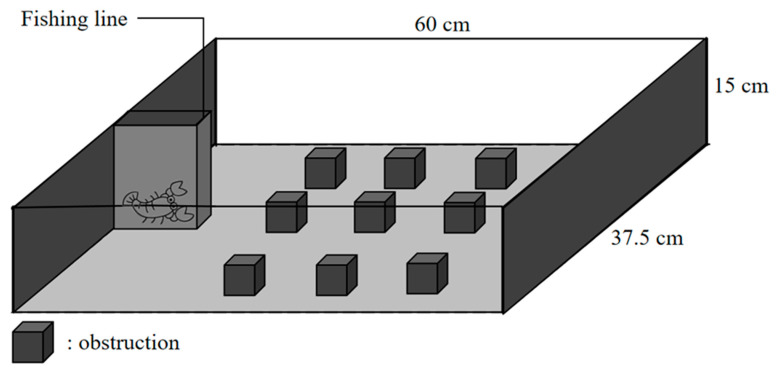
Schematic view of the experimental tank used to measure exploration (lateral view).

**Figure 2 animals-14-01132-f002:**
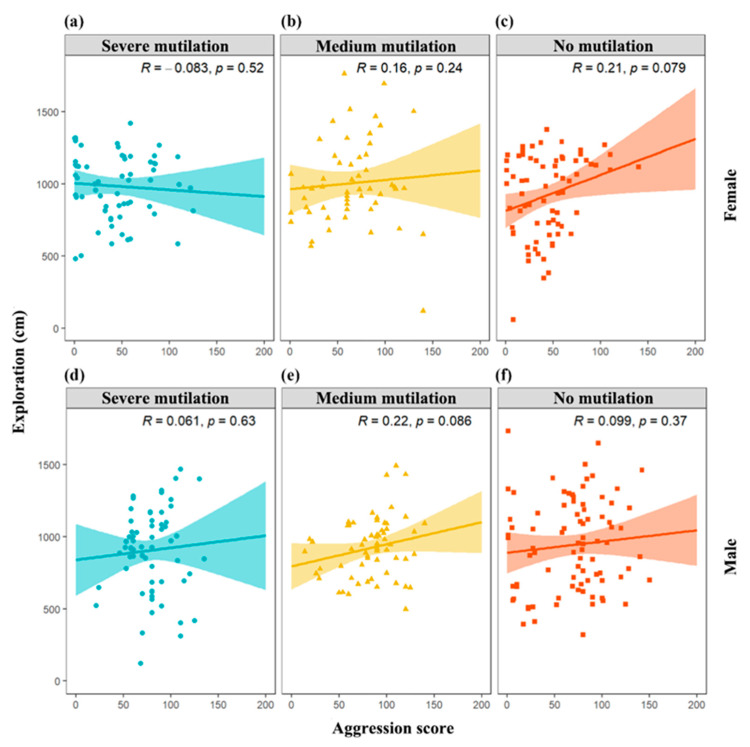
Correlations between aggression and exploration in crayfish with severe mutilation female (**a**), medium mutilation female (**b**), no mutilation female (**c**), severe mutilation male (**d**), medium mutilation male (**e**), and no mutilation male (**f**).

**Figure 3 animals-14-01132-f003:**
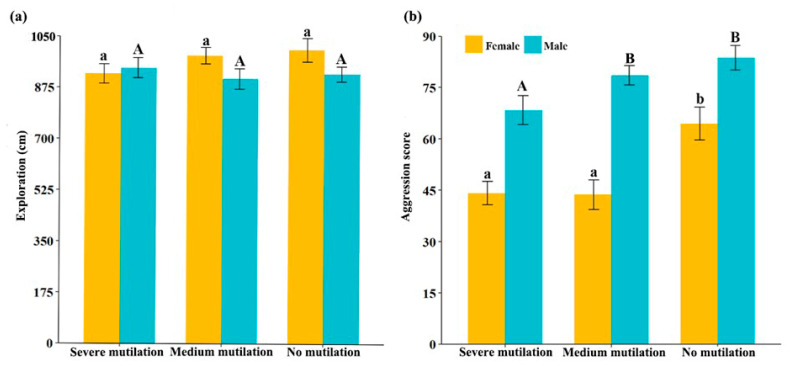
Scores of exploration (**a**) and aggression (**b**) in crayfish with different degrees of cheliped mutilation. Bars with the same letters indicate no significant differences between treatments for each sex.

**Table 1 animals-14-01132-t001:** Mean behavioral traits (±standard deviations (SDs)) of crayfish with different degrees of cheliped mutilation (severe: one cheliped entirely removed; medium: mobile dactyl of one cheliped removed; no mutilation: both chelipeds intact).

Sex	Mutilation Degree	Sample Size	Exploration (cm)	Aggression Score
Female	Severe mutilation	24	922.2 (±57.6)	44.2 (±6.0)
	Medium mutilation	21	984.6 (±49.1)	43.7 (±7.5)
	No mutilation	18	1004.4 (±69.8)	64.5 (±8.3)
Male	Severe mutilation	28	941.9 (±59.6)	68.4 (±7.2)
	Medium mutilation	22	904.7 (±60.3)	78.6 (±4.9)
	No mutilation	21	922.6 (±44.7)	83.8 (±6.2)

**Table 2 animals-14-01132-t002:** Repeatability of exploration in crayfish with different degrees of cheliped mutilation.

Sex	Repeatability	Exploration (cm)		
		Severe Mutilation	Medium Mutilation	No Mutilation
Female	R	0.191	0.18	0.09
	Standard error	0.129	0.13	0.11
	95% confidence interval	0, 0.47	0, 0.45	0, 0.40
	*p*-value	0.06	0.66	0.207
Male	R	0.12	0.38	0.40
	Standard error	0.11	0.14	0.15
	95% confidence interval	0, 0.35	0, 0.59	0.04, 0.61
	*p*-value	0.13	**0.002**	**0.001**

Significantly repeatable behaviors are displayed in bold.

**Table 3 animals-14-01132-t003:** Repeatability of aggression in crayfish with different degrees of cheliped mutilation.

Sex	Repeatability	Aggression Score		
		Severe Mutilation	Medium Mutilation	No Mutilation
Female	R	0.50	0.79	0.57
	Standard error	0.15	0.12	0.17
	95% confidence interval	0.19, 0.73	0.41, 0.88	0.23, 0.83
	*p*-value	**<0.001**	**<0.001**	**<0.001**
Male	R	0.56	0.57	0.96
	Standard error	0.18	0.15	0.02
	95% confidence interval	0.27, 0.89	0.22, 0.78	0.91, 0.98
	*p*-value	**<0.001**	**<0.001**	**<0.001**

Significantly repeatable behaviors are displayed in bold.

## Data Availability

The datasets generated during and/or analyzed during the current study are available from the corresponding author on reasonable request.

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
