# Peer review of "Effects of Cheliped Amputation on the Personality of Crayfish"

_animals, 2024, doi:10.3390/ani14071132_

Round 1

Reviewer 1 Report

Comments and Suggestions for Authors

Line 78: most authors call these projections "hooks", not adapters

Line 101: I'm not sure what you mean to "explore the environment": move around, touch things, wave the antennae? Have you observed such behavior in crayfishes in their stock tanks?

Line 131: Define the size of the crayfish that you call "juveniles": carapace length  from the orbits to the posterior end of the carapace?

Line 170: Why did you use tap water instead of reverse osmosis water? Tap water may be chlorinated.

It would be worthwhile to tell your readers that a male crayfish with only one cheliped almost never is able to court a female, even if the male is large. Females may "explore" as they follow pheromone trails from the male.

Author Response

Response to Reviewer 1 Comments

1. Summary

2. Point-by-point response to Comments and Suggestions for Authors

Comments 1: [Line 78: most authors call these projections "hooks", not adapters]

Response 1: Thanks for your suggestion. We have replaced “adapters” with “hooks”. [this change can be found – page 2, line 79.]

Comments 2: [Line 101: I'm not sure what you mean to "explore the environment": move around, touch things, wave the antennae? Have you observed such behavior in crayfishes in their stock tanks?]

Response 2: We defined exploration as crayfish moving around with antennae and cheliped touching the surroundings. [line 196.]

Comments 3: [Line 131: Define the size of the crayfish that you call "juveniles": carapace length  from the orbits to the posterior end of the carapace?]

Response 3: We identified juveniles as the crayfish with an age of less than 2 months. The size of the "juveniles" crayfish: the length of the carapace from the eye socket to the back end of the carapace is about 4 cm.

Comments 4: [Line 170: Why did you use tap water instead of reverse osmosis water? Tap water may be chlorinated.]

Response 4: The tap water I use has been aerated by sun exposure and oxygen pumps. I change the “tap water” to “dechlorinated tap water”. [line 170.]

Reviewer 2 Report

Comments and Suggestions for Authors

This ms presents a well-conducted study on how alterations to morphology can influence personality in crayfish. I have some suggestions for improvement but none are major. Most concern deletion of redundant words or additional information, and there is a suggestion concerning an alternative explanation that should be explored.

10 and 28 delete "in the animal kingdom"

30 delete "Animal"

94 Change "finger" to "dactyl" also at 127,  246

120 I am not sure what is meant here by "could be neglected" Please rephrase.

120-129 Did any individuals show autotomy when the cheliped was surgically damaged?125 Describe where the cut was made to removed entire cheliped.

148 Did both types of mutilated cheliped regenerate?

251 Delete "Regarding behavioral consistency and correlation"

253 Delete "crayfish" and replace with "experimental groups"

255 Delete "individuals" and replace with experimental groups"

256-7 This is not correct because you did not test for correlations in this context. You tested for differences between experimental groups.

Fig 2 Consider if this should be deleted. It reports only non-significant results.

270 delete "behaviors"

312 You discuss the differences due to mutilation in terms of altered morphology. However, you should consider if the act of mutilation might have caused the difference rather than just the result of mutilation. There are studies that compare animals that have lost chelipeds by different processes. Both groups lack the chelipeds but they stilled differed in behaviour and stress responses because of the manner of treatment.

332 delete " "in crayfish".

This should be a nice paper when improved.

Bob Elwood

Author Response

Response to Reviewer 2 Comments

1. Summary

2. Point-by-point response to Comments and Suggestions for Authors

Comments 1: [10 and 28 delete "in the animal kingdom"]

Response 1: Deleted as suggested. [this change can be found – page 1, line 10 and 28.]

Comments 2: [30 delete "Animal"]

Response 2: Deleted as suggested. [page 1, line 30.]

Comments 3: [94 Change "finger" to "dactyl" also at 127, 246]

Response 3: Deleted as suggested. [line 94,127,246.]

Comments 4: [120 I am not sure what is meant here by "could be neglected" Please rephrase.]

Response 4: Sorry for the confusion. Animal personality might be shaped by genetic backgrounds (Van Oers, et al. 2005). However, the crayfish we used were randomly selected from large breeding ponds rearing newborns given birth by a large number of females. Therefore, the influence of genetic backgrounds might be controlled and could be neglected. We now provide this explanation on Line 119.

Van Oers, Kees, et al. 2005 Contribution of genetics to the study of animal personalities: a review of case studies. Behaviour 142(9-10):1185-1206.

Comments 5: [120-129 Did any individuals show autotomy when the cheliped was surgically damaged?125 Describe where the cut was made to removed entire cheliped.]

Response 5: No crayfish shows autotomy when the cheliped was surgically damaged. One cheliped was entirely removed by cutting at the base. [line 126.]

Comments 6: [148 Did both types of mutilated cheliped regenerate?]

Response 6: For crayfish that had regenerated their cheliped during rearing, the researchers cut them again and disinfected the wounds. [line 149.]

Comments 7: [251 Delete "Regarding behavioral consistency and correlation"]

Response 7: Deleted as suggested. [line 249.]

Comments 8: [253 Delete "crayfish" and replace with "experimental groups"]

Response 8: Deleted as suggested. [line 254.]

Comments 9: [255 Delete "individuals" and replace with experimental groups"]

Response 9: Replaced as suggested. [line 255.]

Comments 10: [256-7 This is not correct because you did not test for correlations in this context. You tested for differences between experimental groups.]

Response 10: We agree with this comment. [line 256-7.]

Comments 11: [Fig 2 Consider if this should be deleted. It reports only non-significant results.]

Response 11: Fig 2 should be retained because it helps to better understand the results.

Comments 12: [270 delete "behaviors"]

Response 12: Deleted as suggested. [this change can be found –line 267.]

Comments 13: [312 You discuss the differences due to mutilation in terms of altered morphology. However, you should consider if the act of mutilation might have caused the difference rather than just the result of mutilation. There are studies that compare animals that have lost chelipeds by different processes. Both groups lack the chelipeds but they stilled differed in behaviour and stress responses because of the manner of treatment.]

Response 13: Yes, how the crayfish loss their chelipeds might influence their behaviours. It is better to have subjects that lost chelipeds not due to experimenters. In our experiment, however, we needed a large number of crayfish with mutilated chelipeds and thus we surgically cut the chelipeds. After the cutting, we observed no abnormal behaviours and the crayfish were continuously reared for a relatively long time (approximated three months) before we measured their behaviours. We believed the long time might eliminate the possible influence of the cutting.

Comments 14: [332 delete " "in crayfish".]

Response 14: Deleted as suggested. [this change can be found –line 328.]

Reviewer 3 Report

Comments and Suggestions for Authors

The paper details the "personality" of crayfish after claw amputation. It is quite a simple experiment and I think the lack of positive results may be due to experimental set-up.

The introduction is very long given that the results and discussion are relatively short - this could be shortened

The authors use the term "personality". I know this is a recognized term but not sure how useful/misleading it is for invertebrates. I think that the word physiotypes would be better. Personalities usually seen in endotherms.

My major concern with the paper is the lack of settling time in the apparatus. Crustaceans remain stressed for 6-12h after handling. Five min is not enough settling time. I think the authors will be recording animals in a stressed and excited state when metabolism is elevated. This will not be natural behaviour and likely will not produce any meaningful results. The measures used are very rudimentary and one could argue a greater diversity of reactions could/should have been followed. 

Comments on the Quality of English Language

OK

Author Response

Response to Reviewer 3 Comments

1. Summary

Thank you very much for taking the time to review this manuscript. Please find the detailed responses below and the corresponding revisions/corrections highlighted/in track changes in the re-submitted files. We also revised the language of the article as suggested.

2. Point-by-point response to Comments and Suggestions for Authors

Comments 1: [The paper details the "personality" of crayfish after claw amputation. It is quite a simple experiment and I think the lack of positive results may be due to experimental set-up.

The introduction is very long given that the results and discussion are relatively short - this could be shortened]

Response 1: I couldn't agree more with you. I shortened this part. [line 57,68,80,87]

Comments 2: [The authors use the term "personality". I know this is a recognized term but not sure how useful/misleading it is for invertebrates. I think that the word physiotypes would be better. Personalities usually seen in endotherms.]

Response 2: I read in the literature and the term "personality" generally applies to invertebrates as well.

1.→Galib, Shams M, et al. 2022 Personality, density and habitat drive the dispersal of invasive crayfish. Scientific Reports 12(1):1114.

2.→Gherardi, Francesca, Laura Aquiloni, and Elena Tricarico 2012 Behavioral plasticity, behavioral syndromes and animal personality in crustacean decapods: An imperfect map is better than no map. Current Zoology 58(4):567-579.

3.→Carere, Claudio, and Francesca Gherardi 2013 Animal personalities matter for biological invasions. Trends in ecology & evolution 28(1):5-6.

Comments 3: [My major concern with the paper is the lack of settling time in the apparatus. Crustaceans remain stressed for 6-12h after handling. Five min is not enough settling time. I think the authors will be recording animals in a stressed and excited state when metabolism is elevated. This will not be natural behaviour and likely will not produce any meaningful results. The measures used are very rudimentary and one could argue a greater diversity of reactions could/should have been followed.]

Response 3: Five minutes of acclimating time was sufficient to transfer them from the feeding tank to the experimental field. The movement of the transfer is slow enough to prevent them from being stimulated and stressed. Our previous study also showed that 5 min of adaptation was sufficient.

1.       Xu, W., Yao, Q., Zhang, W., et al, 2021. Environmental complexity during early life shapes average behavior in adulthood. Behav. Ecol. 32, 105–113.

2.→Si, Mengdi, et al. 2023 Risk Predictability in Early Life Shapes Personality of Mosquitofish in Adulthood. Animals 13(7):1214.
